# Kinetic Analysis of Prostate-Specific Antigen Interaction with Monoclonal Antibodies for Development of a Magnetic Immunoassay Based on Nontransparent Fiber Structures

**DOI:** 10.3390/molecules27228077

**Published:** 2022-11-21

**Authors:** Alexey V. Orlov, Alexandr G. Burenin, Artemiy M. Skirda, Petr I. Nikitin

**Affiliations:** 1Prokhorov General Physics Institute of the Russian Academy of Sciences, 38 Vavilov Street, 119991 Moscow, Russia; 2National Research Nuclear University MEPhI (Moscow Engineering Physics Institute), 31 Kashirskoe Shosse, 115409 Moscow, Russia

**Keywords:** iron oxide nanoparticles, functionalized magnetic labels, nonlinear nanotags, biomolecular interactions, label-free sensing, molecular biomarkers, bioanalytical systems, magnetic immunodetection

## Abstract

Prostate cancer is the second most common cancer diagnosed in men worldwide. Measuring the prostate-specific antigen (PSA) is regarded as essential during prostate cancer screening. Early diagnosis of this disease relapse after radical prostatectomy requires extremely sensitive methods. This research presents an approach to development of an ultrasensitive magnetic sandwich immunoassay, which demonstrates the limit of PSA detection in human serum of 19 pg/mL at a dynamic range exceeding 3.5 orders of concentration. Such attractive performance stems, inter alia, from the kinetic analysis of monoclonal antibodies (mAbs) against free PSA to select the mAbs exhibiting best kinetic characteristics and specificity. The analysis is carried out with a label-free multiplex spectral-correlation interferometry compatible with inexpensive single-use glass sensor chips. The high sensitivity of developed PSA immunoassay is due to electronic quantification of magnetic nanolabels functionalized by the selected mAbs and three-dimension porous filters used as an extended solid phase. The assay is promising for PSA monitoring after radical prostatectomy. The proposed versatile approach can be applied for the rational design of highly sensitive tests for detection of other analytes in many fields, including in vitro diagnostics, veterinary, food safety, etc.

## 1. Introduction

At present, prostate cancer is the second most common cancer diagnosed in men worldwide [1]. Moreover, it is considered to be the cause of about 8% of total cancer mortality [1,2]. Therefore, methods of early diagnosis of this disease are of high demand. Historically, prostate cancer was identified by testing the prostate gland for the presence of indurations on a digital rectal examination or through examining histological specimens of prostate tissue [3,4]. The discovery of prostate-specific antigen (PSA) and its investigation have shown that higher levels of this substance in human seminal plasma may indicate such diseases as prostatitis, benign hyperplasia, and prostate cancer [5,6,7].

Although PSA does not offer very high specificity as a prostate cancer marker [8,9], its concentration in blood along with the ratio of total (tPSA) to free (fPSA) concentrations are now regarded in many countries among the essential parameters to be checked during cancer screening of the male population [10,11]. Moreover, frequent ultrasensitive monitoring of PSA concentration is vital in some cases, e.g., for the patients after radical prostatectomy (RPE), because even minimal PSA concentrations may indicate a disease relapse [12,13]. 

At present, fPSA is determined by various techniques. Many of them, such as assays based on the enzyme-linked immunosorbent assay (ELISA) principle or bead-based chemiluminescent immunoassays, are well developed and have been used for routine medical diagnostics for a long time [14,15]. More recent strategies for high-sensitive quantification of fPSA in human serum [16] employ diverse sensing principles including fluorescence [17,18], luminescence [19], surface-enhanced Raman spectroscopy, localized surface plasmon resonance [20,21], fluorescence resonance energy transfer [22,23], detection of magnetic nanolabels [24,25], etc. At the same time, advanced methods that provide higher sensitivity and specificity are still required, in particular for post-RPE monitoring of PSA [26].

The majority of methods for fPSA determination belong to a class of immunoassays based on antigen-antibody binding. Accordingly, antibody affinity and kinetic parameters of the corresponding biomolecular interaction are among the most important characteristics that determine the assay performance. A multitude of mostly label-free approaches have been proposed for measuring the mentioned parameters, e.g., those based on the surface plasmon resonance, optical resonator-based techniques, isothermal titration calorimetry, and microscale thermophoresis [27,28,29,30,31]. However, a label-free method is still to be developed that uses affordable and inexpensive single-use sensor chips with no precisely deposited thin metal (such as gold) films, e.g., glass slips compatible with well-developed microarrays. 

Here, a label-free method is proposed for investigating the kinetics of biomolecular interaction of fPSA with corresponding antibodies. The method is based on the multiplex spectral correlation interferometry (mSCI) that enables employment of inexpensive single-use glass sensor chips without any metal layers [32,33]. The method is demonstrated by characterization of kinetics of fPSA interactions with different monoclonal antibodies (mAbs). For seven mAbs specific to fPSA, values of the rate and equilibrium association/dissociation constants were determined. Two types of surface modification of the glass sensor chips have enabled investigation of the mAb kinetics with respect to both unlabeled and biotinylated fPSA. The anti-fPSA mAbs selected based on the determined kinetic parameters were used for development of a highly sensitive magnetic immunoassay based on nontransparent fiber 3D solid phases. 

## 2. Materials and Methods

### 2.1. Surface Modification of Microscope Cover Slips and Antibody Immobilization

In this research, cost-efficient single-use sensor chips were made of 100-µm microscope cover glass slips of 22 × 22 mm in size. The chip surface was functionalized by either (i) epoxylation of glass surface with epoxy-functional silane, (3-glycidyloxypropyl)trimethoxysilane (GLYMO; Sigma-Aldrich, St. Louis, MO, USA) (Figure 1a) or (ii) amination of glass surface with amino-functional silane, (3-aminopropyl)triethoxysilane (APTES; Sigma-Aldrich, St. Louis, MO, USA) followed by biotinylation and immobilization of biotinylated monoclonal antibody via a biotin-streptavidin-biotin bridge (Figure 1b). Prior to the surface modification, the glass slips were cleaned as follows. They were first washed with methanol (Sigma-Tech, Moscow, Russia) and kept in a piranha solution for 40 min at 40 °C. Then, the slips were successively washed thrice with triple-distilled water, twice with acetone (Sigma-Tech, Russia), and finally, once with methanol. For the surface epoxylation, the pre-cleaned glass slips were thrice washed with methanol and kept for 16 h in a 5% solution of GLYMO (Sigma-Aldrich, USA) in methanol. After incubation, the slips were kept in a dry-heat oven for 60 min at 105 °C. Then, the slips were washed with dimethyl sulphoxide (DMSO; Sigma-Aldrich, USA) and methanol. The thus modified slips were stored at room temperature. For covalent immobilization of antibody onto the glass surface, a 15 µg/mL solution of monoclonal antibody was pumped for 10 min inside a liquid handling system of the mSCI biosensor. 

For the surface amination, the pre-cleaned glass slips were kept in a mixture of 960 µL of methanol, 10 µL of distilled water, and 30 µL of APTES (Sigma-Aldrich, St. Louis, MO, USA) for 6 h at room temperature. Then, the slips were successively washed with methanol, twice with acetone, and once with distilled water. For covalent immobilization of biotin onto the aminated glass surface, the slips were kept in 5 mM solution of *N*-hydroxysuccinimide ester (NHS-biotin; Sigma-Aldrich, St. Louis, MO, USA) in dimethyl formamide (DMF, Sigma-Aldrich, St. Louis, MO, USA) followed by thrice-washing by DMF and twice by methanol [34]. The obtained biotinylated glass slips were stored at room temperature. The subsequent antibody immobilization onto the glass surface was implemented in two steps inside a liquid handling system of the mSCI biosensor. At first, a 25 µg/mL streptavidin solution in PBS was passed through the flow cell. After that, a solution of biotinylated antibody in PBS (at concentration of 15 µg/mL) was passed, followed by blocking with 0.1% bovine serum albumin (BSA, Sigma-Aldrich, St. Louis, MO, USA) in PBS. The results of step-by-step characterization of the surface modification are shown in Appendix A.

### 2.2. Label-Free Registration of Association/Dissociation of Antibody-Antigen Complexes

The sensor chips prepared as described in Section 2.1 were employed for mSPI-based assessment of kinetic characteristics (rate and equilibrium constants of association/dissociation) of the unlabeled or biotinylated mAb immobilized on the chips. 

The mSCI technique is based on direct optical sensitive registration of changes of biolayer thickness, which varies during antibody-antigen binding (Figure 1c). The mSCI principle allows using widely available cover glass slips as the sensor chips [33,35,36]. Seven different anti-fPSA mAb clones were studied, namely: M612165, M612166 (Fitzgerald Industries, Acton, MA, USA); ICO204, ICO168, 3F8, 4H3, 2/2C2 (Alkorbio, St. Petersburg, Russia). A 5 µg/mL solution of fPSA in a blocking buffer was pumped along the sensor chip surface by a peristaltic pump at 5 µL/min for 25 min to register association process. Then, the blocking buffer without fPSA was pumped for 3 h for recording dissociation. 

### 2.3. Magnetic Immunoassay

The magnetic immunoassay for fPSA detection has been developed using the mAb that showed the best kinetic characteristics. The assay is based on porous volumetric structures as a solid phase, 50-nm magnetic nanoparticles functionalized by streptavidin (Miltenyi Biotec, Bergisch Gladbach, Germany), and their electronic registration by nonlinear remagnetization at combinatorial frequencies with the method of magnetic particle quantification (MPQ) [37,38,39]. 

The solid phase in the assay uses filters made of polyethylene fibers coved by a polypropylene layer. Each filter is a 5-mm long cylinder of 3 mm in diameter. The filters were preliminary soaked in 96% ethanol for 5 min, then in 50% ethanol for 5 min followed by twice-washing with a carbonate bicarbonate buffer (pH 9.6). After that, the filters were placed into a 40 µg/mL solution of unlabeled antibody for 3 h at room temperature. Then, the filters were washed two times by a 0.1% solution of Tween-20 in PBS (PBST) and kept in a 5% solution of BSA in PBS for 1 h. After the incubation, the filters were twice washed in PBST and put into tips compatible with an automatic pipette used for the immunoassay.

The calibration serum samples of concentrations of 0, 0.03, 0.1, 0.3, 1, 3, 10, 30, 100 ng/mL were prepared by spiking fPSA (Alkorbio, Russia) into PSA-free female serum. 

The immunoassay procedure was as follows. The calibration serum samples were pumped for 10 min through the filters, on which unlabeled mAbs were immobilized, followed by washing with PBST. Then, a 15 µg/mL solution of biotinylated antibody in PBS was passed through tips for 7 min with subsequent washing by PBST. After that, a solution containing magnetic particles functionalized with streptavidin was passed for 5 min followed by PBST washing. Eventually, the filters were placed into an MPQ reader to read out the magnetic signal. 

### 2.4. Data Processing

Each experiment was implemented at least three times. The data are presented as mean values with respective standard deviations. The fragments of temporal dependences of biolayer thickness (sensorgrams), which corresponded to the biomolecule association process, were fitted by the function:(1)$$R\left(t\right)=R_{max}\cdot \left(1-exp^{-\left(C\cdot k_{on}+k_{off}\right)\cdot t}\right),$$
where $R_{max}$—maximal signal value, C—analyte concentration, $k_{on}$ and $k_{off}$—constants of reaction rates of association and dissociation, respectively [40,41]. The sensorgram fragments that corresponded to the biomolecule dissociation were fitted by the function:(2)$$R\left(t\right)=R_{max}\cdot \left(exp^{-k_{off}\cdot t}\right).$$

The analytical limit of detection (LOD) of the magnetic immunoassay was determined as the fPSA concentration on a calibration curve, at which the magnetic signal differs from the signal of zero-concentration sample by two standard deviations [42].

## 3. Results and Discussion

### 3.1. Kinetic Characterization of Unlabeled Anti-fPSA Antibodies Covalently Immobilized on Epoxylated Sensor Chips

Figure 2 exhibits an experimental setup for determining kinetic characteristics of fPSA with unlabeled antibody and the related characteristic sensorgrams obtained using the multiplex spectral-correlation interferometry. In this setup, we used the epoxylated-surface sensor chips with different anti-fPSA mAb clones immobilized in different spots on the chips. During pumping along the chip surface of samples, which contained spiked fPSA, the antibody association with the respective antigen and the immunocomplex formation were recorded in real time. Then, while pumping a sample without fPSA, the immunocomplex dissociation was registered. Eventually, rate and equilibrium constants of association and dissociation were calculated.

The calculated values of rate and equilibrium constants of association and dissociation for each of the seven studied clones of anti-fPSA mAbs, namely M612165, M612166, ICO204, ICO168, 3F8, 4H3, 2/2C2, are shown in Table 1. As can be seen, 4H3 clone features the best rate-constant of association and the best equilibrium constants. Such clone will facilitate faster antibody-antigen binding along with faster approach to equilibrium and most efficient binding with fPSA at low concentrations. Therefore, this antibody proves to be the most promising for development of express methods of sensitive registration of fPSA. Interestingly, the best value of dissociation constant was demonstrated by the antibody of ICO204 clone. This indicates the tightest bond of this antibody with fPSA in the immunocomplex, which would offer the slowest dissociation. That factor may be of key importance for development of lengthy multi-stage assays or, e.g., for sorption on immunoaffinity columns. 

For the M612166 clone, one may find, in literature, slightly different values of constants but of the same order of magnitude measured with another label-free technique of surface plasmon resonance (SPR) [43]. Katsamba et al. [43] also used covalent immobilization of mAbs but substantially dissimilar sensor chips and their surface, namely carboxymethyldextran deposited onto a thin gold film to be activated by carbodiimide. The SPR-measured values of K_d_ and K_a_ were 1.1 × 10^−9^ M and 9.1 × 10^8^ M^−1^, respectively, whereas those obtained with our mSCI method were 7.1 × 10^−10^ M and 1.4 × 10^9^ M^−1^. The slight discrepancy in the values may be due to the mentioned differences in the sensing surface types, which may result in different activity of the immobilized antibody molecules.

The absence of non-specific contribution caused by fPSA binding directly to the slide rather than to the antibody was verified in control experiments using the glass sensor chips without immobilized antibody to fPSA. Instead of antibodies, in these experiments we pumped blocking buffer along the sensor chip followed by passing fPSA antigen. No noticeable non-specific binding of fPSA with the glass surface was registered (Appendix A).

An important advantage of our registration technique is its capacity of multiplex operation for independent and simultaneous label-free measuring of several analytes using multi-spot sensor chips. Another benefit is wide availability and cost-efficiency of single-use consumables because mSCI allows using standard glass cover slips as the sensor chips, each registration spot on which being independently read by a CCD-detector.

### 3.2. Kinetic Characterization of Biotinylated Antibodies

One of the versatile approaches to enhancing the sensitivity of a fPSA test system is using biotin-labeled anti-fPSA mAbs so that the ultra-high affinity of biotin-streptavidin interaction (Kd) can be employed for, e.g., signal amplification or more efficient sorption of antibody molecules on the surface [44,45]. The biotinylation may affect the observed interaction constants, for instance, because of introduction of the biotin tag directly to the area of antigen-recognizing site of antibody or in its close vicinity [46,47]. Besides, at a non-oriented immobilization of the biotinylated antibodies, the observed affinity may change due to an unalike profile of hindering the antigen-binding sites [48]. Accordingly, the kinetic parameters of interaction of biotinylated antibodies with fPSA is of particular interest. 

For investigation of the biotinylated anti-fPSA mAbs, we used a setup similar to that described in Section 3.1 except we applied a sensor chip with a biotinylated surface onto which we first sorbed streptavidin and then immobilized biotinylated antibodies to form a stable “biotin—streptavidin—biotinylated mAbs” complex (Figure 3). To determine the rate and equilibrium constants shown in Table 2, a solution containing spiked fPSA was passed, followed by pumping a solution without fPSA. 

It can be seen from the comparison of Table 1 and Table 2 that the same pair of antibody clones (4H3 and ICO168) exhibits the best kinetic characteristics in both unlabeled and biotinylated setups, with 4H3 antibody showing better characteristics in the unlabeled variant and ICO168 in the biotinylated one. It should be noted that for ICO168 clone, the difference in kinetic characteristics of the unlabeled and biotinylated antibodies is within the experimental error. Meanwhile, for some other clones, such as ICO204 and 2/2C2, this difference is significant—up to 1–2 orders of magnitude. For all clones except 2/2C2, the rate constants of dissociation do not change within the experimental error for the unlabeled and biotinylated antibodies. That may indicate that 2/2C2 is the only clone whose kinetic variations are caused by, inter alia, the fact that its biotinylation has considerably affected antigen-binding sites.

A separate experimental study was devoted to specificity characterization of both unlabeled and biotinylated antibodies. In these experiments, instead of PSA, we pumped solutions of the following non-target antigens in concentrations exceeding the physiological ranges: carcinoembryonic antigen (CEA), 100 ng/mL; alpha-fetoprotein (AFP), 300 ng/mL; ovarian cancer-related tumor marker CA125, 150 U/mL; thyroid stimulating hormone (TSH), 300 mIU/L; triiodothyronine (T3), 5 ng/mL; thyroxine (T4), 200 pM. No noticeable changes in the biolayer thickness were registered at the stage of antigen binding (see Appendix A). Thus, the studied antibodies demonstrated high specificity, probably because they were of commercial grade, and their manufacturers paid particular attention to selection of highly specific clones.

The two clones that exhibited the best kinetic characteristics, namely unlabeled 4H3 and biotinylated ICO168, were further used for development of a highly sensitive label-based sandwich immunoassay for fPSA determination. These clones were pre-tested with the label-free method of mSCI to verify their effectiveness in forming the “antibody—antigen—antibody” immune sandwich (see Appendix A). Such verification is very important as two antibody clones with good kinetic parameters often cannot be paired, for example, if they recognize overlapping epitopes or compete for the same binding site. Thus, the proposed approach based on mSCI employment has enabled implementation of a complete cycle of investigation and characterization of the antibodies to be further used in the development of sandwich-type analytical systems.

### 3.3. Development of Highly-Sensitive Immunoassay for fPSA Detection

The highly sensitive immunoassay for fPSA detection developed in this research is based on magnetic nanoparticles used as the labels which are electronically registered by non-linear remagnetization with the MPQ technique. This registration technique is based on subjecting a sample to a magnetic field of two frequencies f_1_ = 100 Hz and f_2_ = 100 kHz with signal recording at f_2_ ± 2f_1_ (Figure 4a) [37,38]. In the assay, three-dimensional porous filters of polyethylene fibers covered by polypropylene, which were described in detail in Ref. [49], were used as a solid phase for rapid and highly sensitive fPSA quantification. The unlabeled 4H3 antibody was immobilized onto the filter surface, and then the following solutions were sequentially passed along the surface: (i) analyte; (ii) biotinylated ICO168 antibody; (iii) 50-nm magnetic nanoparticles with streptavidin molecules immobilized on their surface (Figure 4b).

The total magnetic assay time is 25 min, which is much less than that of traditional ELISA (2–4 h). The calibration plot of the developed magnetic label-based assay for fPSA detection is presented in Figure 4c as a dependence of magnetic signal registered by the MPQ reader upon fPSA concentration in calibration samples of human serum. The calibration samples were prepared by spiking fPSA into PSA-free female serum in the following concentrations: 0.01, 0.03, 0.1, 0.3, 1, 3, 7.5, 15, 30, 50, 75, and 100 ng/mL. The analytical limit of detection determined by 2σ criterion was 19 pg/mL at a dynamic range of more than 3.5 orders of magnitude. Notably, the reference range of fPSA is calculated as a ratio to the total PSA, and equals 25–100%. The total PSA in healthy patients ranges from 0 to 4 ng/mL. Therefore the highest allowable concentration of fPSA in serum is 1–4 ng/mL, and LOD for the methods for monitoring after the radical prostatectomy should be as low as reasonably achievable [50,51,52,53,54,55,56,57,58]. Thus, the developed assay covers the whole range of physiologically relevant fPSA concentrations in human serum.

Additionally, the specificity of the developed assay, as well as the absence of false-positive signals in the presence of non-target analytes were demonstrated similarly to the procedure described in Section 3.2 (Figure 4d). Thus, the analytical characteristics of the developed 25-min magnetic sandwich immunoassay are better than those of a majority of test systems (Table 3). The assay offers high sensitivity, wide linear range, and short assay time. Notably, the magnetic nanolabels ensure the long-term stability of the recorded signals: the difference between the values registered immediately and 2 months after the assay is within the error (Appendix A).

## 4. Conclusions

The proposed concept of a label-free kinetic analysis of anti-fPSA mAbs with multiplex spectral-correlation interferometry (mSCI) has enabled us to (i) determine the rate and equilibrium constants for 7 different antibodies both in unlabeled and biotinylated forms; (ii) carry out their comparative kinetic characterization and assess their specificity; and (iii) select the most efficient antibodies in view of an immunoassay based on a sandwich of “unlabeled mAb—fPSA—biotinylated mAb—magnetic particle functionalized with streptavidin.” As a result, the selected antibodies were used in the development of a sandwich magnetic immunoassay for highly sensitive detection of the free form of prostate-specific antigen with electronic MPQ-based registration. The achieved limit of fPSA detection is 19 pg/mL, and the dynamic range exceeds 3.5 orders.

Future improvements of the developed magnetic immunoassay, which is promising for post-radical prostatectomy diagnostics, may include clinical trials for thorough investigation of its clinical sensitivity and specificity. The proposed versatile approach to immunoassay development can, in the future, be applied for the rational design of highly sensitive test systems for other important analytes in different areas of in vitro diagnostics, from veterinary to ecological monitoring and food inspections, including express tests based on magnetic lateral-flow principles [59,60]. Additionally, the proposed concept can be extended for characterization of other recognizing biomolecules (such as aptamers, nanobodies, etc.) to enable further improvements in design of high selective and sensitive analytical systems.

## Figures and Tables

**Figure 1 molecules-27-08077-f001:**
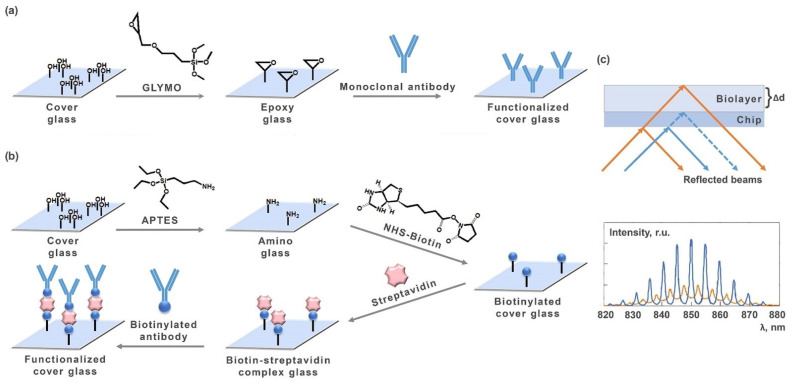
Schemes of modification of microscope cover slip surface for (**a**) covalent immobilization of unlabeled PSA (epoxy-modified glass surface) and (**b**) immobilization onto the biotinylated surface of a biotinylated antibody by means of a streptavidin bridge; (**c**) scheme of SCI-based direct optical sensitive registration of changes of biolayer thickness.

**Figure 2 molecules-27-08077-f002:**
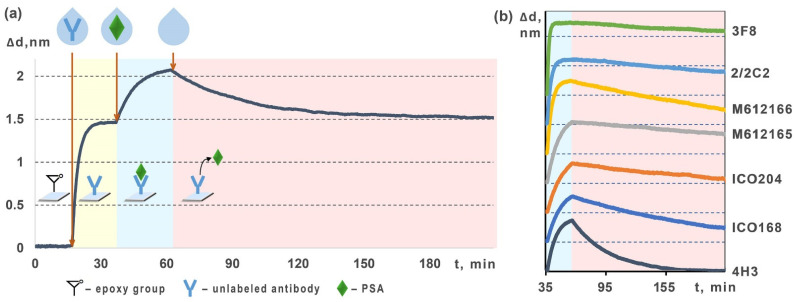
Scheme for determining kinetic characteristics of fPSA interaction with unlabeled antibody: (**a**) characteristic sensorgram obtained with the mSCI that shows all stages of the assay; and (**b**) fragments correspondent to the association and dissociation stages in the sensorgrams registered for seven different clones of anti-fPSA mAbs.

**Figure 3 molecules-27-08077-f003:**
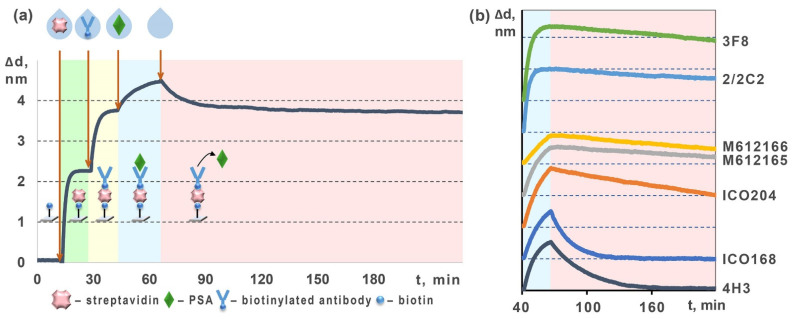
Schematic of label-free determination of kinetic characteristics of fPSA interaction with biotinylated antibodies: (**a**) characteristic mSCI-recorded sensorgram that shows all assay stages; (**b**) sensorgram fragments that show the association–dissociation stages for seven different anti-fPSA mAbs clones.

**Figure 4 molecules-27-08077-f004:**
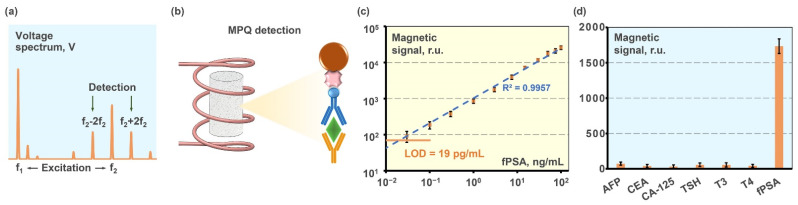
Magnetic label-based determination of fPSA: (**a**) MPQ principle; (**b**) scheme of magnetic assay; (**c**) calibration plot; (**d**) demonstration of assay specificity.

**Table 1 molecules-27-08077-t001:** Values of rate and equilibrium constants of association and dissociation observed for different clones of anti-fPSA mAbs measured on a sensor chip with epoxylated surface.

Antibody Clone	*k_on_*, M^−1^s^−1^	*k_off_*, s^−1^	K_a_, M^−1^	K_d_, M
3F8	6.94 × 10^6^	2.75 × 10^−2^	2.52 × 10^8^	3.96 × 10^−9^
2/2C2	5.29 × 10^6^	7.50 × 10^−3^	7.05 × 10^8^	1.42 × 10^−9^
M612166	4.56 × 10^6^	3.24 × 10^−3^	1.41 × 10^9^	7.11 × 10^−10^
M612165	6.49 × 10^6^	1.51 × 10^−3^	4.29 × 10^9^	2.33 × 10^−10^
ICO204	1.70 × 10^6^	3.40 × 10^−3^	4.99 × 10^9^	2.00 × 10^−10^
ICO168	2.13 × 10^7^	1.35 × 10^−3^	1.58 × 10^10^	6.32 × 10^−11^
4H3	5.10 × 10^7^	9.95 × 10^−3^	5.13 × 10^10^	1.95 × 10^−11^

**Table 2 molecules-27-08077-t002:** Values of rate and equilibrium constants of association and dissociation observed for different clones of anti-fPSA biotinylated mAbs measured on a sensor chip with a biotinylated surface.

Antibody Clone	*k_on_*, M^−1^s^−1^	*k_off_*, s^−1^	K_a_, M^−1^	K_d_, M
3F8	5.06 × 10^6^	3.16 × 10^−2^	1.60 × 10^8^	6.24 × 10^−9^
2/2C2	1.16 × 10^5^	5.90 × 10^−2^	1.96 × 10^6^	5.09 × 10^−7^
M612166	4.13 × 10^6^	3.69 × 10^−3^	1.12 × 10^9^	8.95 × 10^−10^
M612165	4.95 × 10^6^	1.40 × 10^−3^	3.53 × 10^9^	2.83 × 10^−10^
ICO204	1.22 × 10^6^	3.15 × 10^−3^	3.88 × 10^8^	2.58 × 10^−9^
ICO168	2.04 × 10^7^	1.38 × 10^−3^	1.48 × 10^10^	6.76 × 10^−11^
4H3	1.46 × 10^7^	1.12 × 10^−3^	1.30 × 10^10^	7.67 × 10^−11^

**Table 3 molecules-27-08077-t003:** Comparison of different methods for fPSA detection.

Detection Method	Dynamic Range	Assay Time	LOD	Ref.
Electrochemical	1–30	1 h	1 ng/mL	[14]
Fluorescence	3.4–34	>8 h	1.7 ng/mL	[17]
Fluorescence	1–400	1.5 h	25 ng/mL	[18]
Localized surface plasmon resonance	0.2–1	N/A	0.2 ng/mL	[20]
Fluorescence resonance energy transfer	0.005–10	4 h	0.95 pg/mL	[22]
Magnetic particle quantification	0.03–100	25 min	19 pg/mL	This work
Giant magnetoresistance	0.1–50	15 min	70 pg/mL	[24]
Electrochemical (magnetic force-assisted)	0.085–30	5 min	85 pg/mL	[25]

## Data Availability

Not applicable.

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
