# Peer review of "Kinetic Analysis of Prostate-Specific Antigen Interaction with Monoclonal Antibodies for Development of a Magnetic Immunoassay Based on Nontransparent Fiber Structures"

_molecules, 2022, doi:10.3390/molecules27228077_

Round 1

Reviewer 1 Report

In this paper authors showed the use of label-free sensor for the specific detection of PSA a marker for prostate cancer. The sensor find application also in different field and with different cancer markers making them really interesting. The work is well design and clear and the sensor specificity is high. I was wondering if there might be a non-specific contribution made by PSA binding directly to the slide and not to the antibody. Did the authors test this? I suggest to revise the manuscript for minor errors and typos.  I suggest the publication after few comments that should be addressed:

1-I was wondering if there might be a non-specific contribution made by PSA binding directly to the slide and not to the antibody. Did the authors test this?
2-I suggest to revise the manuscript for minor errors and typos.
3-An intruductive figure showing the mechanism of the sensor and the magnetic measurments should be added to the introduction. 

Reviewer 2 Report

This research presents an approach to development of an ultrasensitive magnetic sandwich immunoassay, which demonstrates the limit of PSA detection in human serum of 19 pg/mL at a dynamic 15 range exceeding 3.5 orders of concentration. Overall, the assay is promising for PSA monitoring after radical prostatectomy. I recommend this manuscript to be accepted after replying to the following questions.

1. The observation time of Figure 1 and Figure 2 should be consistent. For example, the monitoring period need to be extended in 120 min in Figure 1a and Figure 2a.

2. The value of R2 should be provided for Figure 4b. Long-term storage stability of such label-free detection method also needs to be studied.

3. The step-by-step surface modification should be characterized through FTIR and contact angle.

4. The limit of detection of PSA from previous studies need to be summarize into a new table for reference purpose.

5. Authors need to be clearly indicate the future improvement to be done on the basis of current paradigm.  

Reviewer 3 Report

The authors have raised very important topic, concering prostate cancer, which is the second most common cancer in men worldwide. They developed a kinetic analysis of monoclonal antibodies (mABs) against free prostate-specific antigen (PSA), which as they elucidated is promising for post-radical prostatectomy diagnostics.

The obtained results are promising and the present work brings a new and innovative strategy
in prostate cancer diagnostics, especially as was here mentioned in monitoring disease relapse. The method presented can be also used to develop high-sensitivity tests used in other fields.

This is extremely important because PSA does not show high specificity as a marker of prostate cancer, and is recommended as a screening parameter in routine testing of the male population.

Round 2

Reviewer 2 Report

The reviewer's doubts are cleared.  Acceptance is recommended.